# The GIRK1 subunit potentiates G protein activation of cardiac GIRK1/4 hetero-tetramers

**Kouki K Touhara, Weiwei Wang, Roderick MacKinnon***

Laboratory of Molecular Neurobiology and Biophysics, Howard Hughes Medical Institute, Rockefeller University, New York, United States

**Abstract** G protein gated inward rectifier potassium (GIRK) channels are gated by direct binding of G protein beta-gamma subunits (Gβγ), signaling lipids, and intracellular $Na^+$. In cardiac pacemaker cells, hetero-tetramer GIRK1/4 channels and homo-tetramer GIRK4 channels play a central role in parasympathetic slowing of heart rate. It is known that the $Na^+$ binding site of the GIRK1 subunit is defective, but the functional difference between GIRK1/4 hetero-tetramers and GIRK4 homo-tetramers remains unclear. Here, using purified proteins and the lipid bilayer system, we characterize Gβγ and $Na^+$ regulation of GIRK1/4 hetero-tetramers and GIRK4 homo-tetramers. We find in GIRK4 homo-tetramers that $Na^+$ binding increases Gβγ affinity and thereby increases the GIRK4 responsiveness to G protein stimulation. GIRK1/4 hetero-tetramers are not activated by $Na^+$, but rather are in a permanent state of high responsiveness to Gβγ, suggesting that the GIRK1 subunit functions like a GIRK4 subunit with $Na^+$ permanently bound.

## Introduction

In the cardiovascular system, cardiac GIRK channels play a central role in parasympathetic slowing of the heart. Specifically, when the body is at rest, parasympathetic neurons convey signals from the central nervous system to cardiac pacemaker cells *via* cholinergic neurotransmission, activating the muscarinic acetylcholine receptor 2 (M2R). Activated M2Rs release inhibitory G protein alpha subunits and Gβγ. Gβγ is a hetero-dimeric protein composed of tightly bound beta and gamma subunits. This free Gβγ, along with its lipid anchor, diffuses on the intracellular membrane surface and binds directly to GIRK to activate it (*Logothetis et al., 1987*; *Whorton and MacKinnon, 2013*; *Sakmann et al., 1983*; *Kurachi et al., 1986*). Activation of GIRK shifts the resting membrane potential of pacemaker cells toward the equilibrium potential for $K^+$, lengthening the interval between cardiac action potentials and thereby slowing the heart (*Loewi and Navratil, 1926*; *Rayner and Weatherall M, 1959*). The critical role of parasympathetic regulation of cardiac GIRK channels is evident from the severe diseases that result from mutations in the *GIRK* gene such as Atrial Fibrillation (*Kovoor et al., 2001*; *Voigt et al., 2010*), and Long QT syndrome (*Yang et al., 2010*).

Mammals express four GIRK channel subunits (GIRK1-4), forming various homo-tetramers and hetero-tetramers. Cardiac GIRK channels are composed of GIRK1 and GIRK4 subunits (*Krapavinsky et al., 1995*). Since the GIRK1 subunit does not form functional homo-tetramers, GIRK1 and GIRK4 subunits form functional GIRK1/4 hetero-tetramers and GIRK4 homo-tetramers in the heart (*Krapavinsky et al., 1995*; *Chan et al., 1996*; *Corey and Clapham, 1998*). *GIRK1* and *GIRK4* knockout mice show similar phenotypes in terms of heart rate (*Bettahi et al., 2002*), suggesting that both subunits perform non-redundant tasks. However, little is known about whether or how GIRK1 influences cardiac GIRK channel behavior. Specifically, what are the functional differences between GIRK1/4 hetero-tetramers and GIRK4 homo-tetramers?

**\*For correspondence:** mackinn@rockefeller.edu

**Competing interests:** The authors declare that no competing interests exist.

**eLife digest** Signals from outside of a cell can alter the activity inside the cell. This process often involves members of a large family of proteins called G protein-coupled receptors (GPCRs) that are found on the surface of many cells in the body. When these receptors are activated they release a G protein on the inside of the cell that then splits into two parts. One of these parts – called the Gβγ subunit – can directly bind to, and open, a protein called a GIRK channel that is found in the cell's membrane. Once opened, these channels allow potassium ions to flow into the cell.

GIRK channels are involved in a number of processes in the body. For example, when we are at rest, our brain sends nerve impulses to the heart and, via signals through GPCRs and Gβγ subunits, causes the GIRK channels to open. The flow of potassium into the heart muscle cells then helps to slow the heart rate.Heart cells produce two subtypes of GIRK channels, called GIRK4 and GIRK1/4. However, it was not known whether these two channels play similar or distinct roles.

Now, Touhara et al. report that GIRK4 and GIRK1/4 channels are distinct. In particular, the way that GIRK4 channels respond to the stimulation from the nervous system can be tuned by the concentration of sodium ions inside the heart cell. When there are more sodium ions in the cell, the Gβγ subunits bind more strongly to the GIRK4 channel; this means that the channel is more sensitive to the nerve impulses from the brain. In a related study, Wang et al. – who include all of the same researchers – also discovered that sodium ions affect GIRK2 channels from neurons in a similar way. By contrast, Touhara et al. found that the GIRK1/4 channel is unaffected by the sodium level inside the cell and is instead always sensitive to stimulation by nerve impulses that signal being at rest.

Touhara et al. then looked at mouse heart cells that had been grown in the laboratory and found that they respond as if all of their GIRK channels were the GIRK1/4 type. That is to say, that a heart cell's activity didn't change much when extra sodium ions were present. This is likely because, unlike in neurons, the concentration of sodium ions inside a heart cell probably does not change much under normal conditions.

These findings shed new light on G protein signaling, but there is still more that is not yet completely understood. For example, different GPCRs in cells will all release Gβγ subunits when stimulated but somehow produce specific responses. Touhara et al. are now interested in figuring out how this kind of specificity is achieved in heart cells.

Although GIRK1 and GIRK4 subunits share $\sim$44% sequence identity, one notable difference occurs in the $Na^+$ binding site. The GIRK1 subunit has an aspartate to asparagine replacement in this $Na^+$ binding site, presumably rendering it incapable of binding intracellular $Na^+$ (*Ho and Murrell-Lagnado, 1999*). However, it is still unclear what influence this defective $Na^+$ binding site has on the function of GIRK1/4 hetero-tetramers. Cellular electrophysiological experiments have not clarified this issue because it is difficult to control the concentration of GIRK ligands inside cells and it is also not possible to express GIRK1/4 hetero-tetramers without co-expression of GIRK4 homo-tetramers. To overcome these difficulties we have purified human GIRK1/4 hetero-tetramers and GIRK4 homo-tetramers and studied their ligand regulation by $Na^+$ and Gβγ in the planar lipid bilayer system.

## Results

### Purified GIRK1/4 hetero-tetramers and GIRK4 homo-tetramers are functional in planar lipid bilayer membranes

Although the GIRK1 subunit does not form functional homo-tetrameric channels, it does form structural homo-tetramers similar to GIRK4 (*Figure 1*). Therefore, in order to isolate GIRK1/4 hetero-tetramers, GIRK1 and GIRK4 homo-tetramers had to be removed during purification. To remove both homo-tetramers two different tags, a deca-histidine tag and a 1D4 peptide tag, were fused to the GIRK1 and GIRK4 subunits, respectively. Two sequential affinity chromatography steps isolated only GIRK1/4 hetero-tetramer channels containing both tags (*Figure 2A*). Equal bands in all lanes of an SDS-PAGE gel, corresponding to different elution fractions from a gel-filtration column, suggested that the predominant channel species purified contained two GIRK1 and two GIRK4 subunits

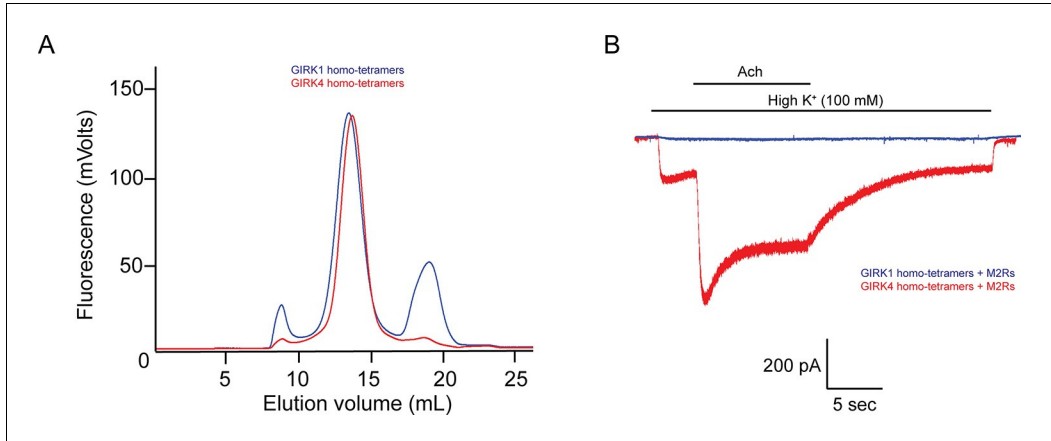

**Figure 1.** The GIRK4 subunit forms functional homo-tetrameric channels, whereas the GIRK1 subunit forms nonfunctional homo-tetramers. (**A**) HEK293T cells were transiently transfected with the GIRK1 or the GIRK4 subunit fused to GFP, and solubilized cell lysate was analyzed by fluorescent size-exclusion chromatography (Superose 6 10/300 GL). Blue and red elution profiles show GIRK1 homo-tetramers and GIRK4 homo-tetramers, respectively. (**B**) HEK293T cells were transiently transfected with GIRK1 (blue) or GIRK4 (red), and human M2Rs. Whole-cell voltage clamp recordings were performed. Membrane potential was held at -80 mV, and the extracellular solution was exchanged to high potassium buffer (100 mM KCl) as indicated above the signal, followed by the application of 10 μM acetylcholine.

(*Figure 2B*). This suggestion is based on the different elution times of homo-tetramer GIRK1 and GIRK4 subunits (*Figure 1A*). We cannot, however, exclude with certainty the possibility that some channels with 3:1 and/or 1:3 stoichiometry were present in the population of isolated channels.

Purified GIRK channels were reconstituted into liposomes and fused with planar lipid bilayer membranes. The channels were activated by fusing lipid-anchored Gβγ-containing vesicles with the membranes and adding the membrane-impermeable, short-chain $PIP_2$ (C8-$PIP_2$) to one chamber of the planar bilayer. Although channels and Gβγ insert into the bilayer membrane randomly in both orientations, only channels with their intracellular surface facing the chamber to which $PIP_2$ was added are activated (*Wang et al., 2014*). The strong inward-rectification of current as a function of membrane voltage supports the uniform orientation of active channels (*Figure 2C*).

In contrast to GIRK1 homo-tetramers, GIRK4 homo-tetramers form functional channels that are activated by GPCR stimulation when expressed in HEK293T cells (*Figure 1B*). To nullify any residual uncertainty that GIRK4 may actually form functional channels in cells by combining with native GIRK1 subunits that may be present, we purified and reconstituted GIRK4 homo-tetramers and found they produce robust inward-rectifier $K^+$ currents in planar lipid membranes (*Figure 2D*).

## The Na$^+$-insensitive GIRK1 subunit potentiates Gβγ activation of GIRK1/4 hetero-tetramers

To study the dependence of GIRK channel activity on $Na^+$ and Gβγ concentrations, we used lipids with Ni-NTA modified head groups (Ni-NTA-lipids) as illustrated (*Figure 3A*) using a method described in the accompanying paper (*Wang et al., 2016*). In this method, bilayer membranes containing specific mole fractions of Ni-NTA-lipids were formed. GIRK channel proteoliposomes, which also contained the same mole fraction of Ni-NTA-lipids as the bilayer membrane, were then fused to the membrane. C8-$PIP_2$ and 2 μM soluble Gβγ (sGβγ-His$_{10}$), which contained a deca-histidine-tag instead of its physiological lipid anchor, were applied to the intracellular side of the membrane. At 2 μM concentration sGβγ-His$_{10}$ does not activate GIRK channels directly from solution, however, it saturates (i.e. occupies nearly 100% of) all available Ni-NTA-lipids in the membrane (*Wang et al., 2016*). These membrane-bound sGβγ-His$_{10}$ molecules are able to activate GIRK channels, which are present in the membrane at a much lower density than Ni-NTA-lipid molecules (*Figure 3*). This method permits the study of GIRK channel activation as a function of the membrane sGβγ-His$_{10}$ density (Gβγ concentration), which is controlled through the predetermined mole fraction of Ni-NTA-

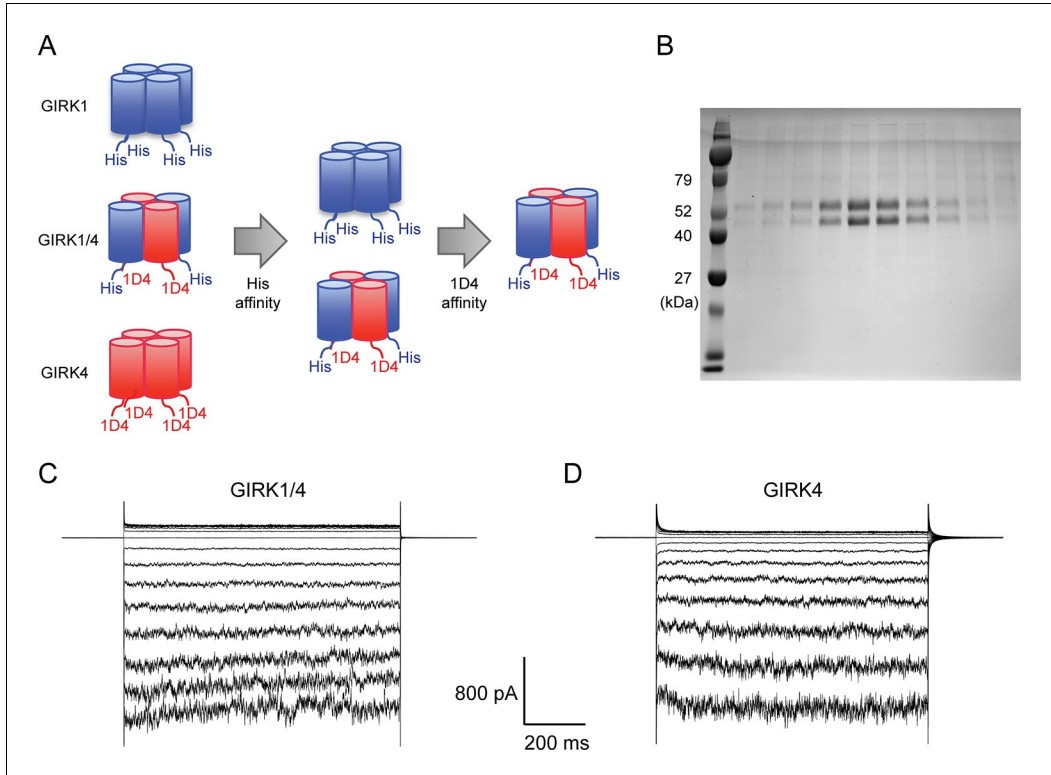

**Figure 2.** Purified cardiac GIRK channels are functional in reconstituted planar lipid bilayer membranes. (**A**) Schematic of GIRK1/4 hetero-tetramer purification procedure. 1D4-tagged GIRK4 homo-tetramers were removed with $Co^{2+}$ affinity chromatography and His-tagged GIRK1 homo-tetramers were removed with subsequent 1D4 affinity chromatography. (**B**) Gel-filtration fractions of the GIRK1/4 hetero-tetramer peak were run on 12% SDS-PAGE. GIRK1 and GIRK4 monomers are 56 kDa and 46 kDa, respectively. (**C**) and (**D**) The top and bottom chambers are separated by the lipid bilayer formed on a transparency film. The same solution containing 10 mM potassium phosphate buffer pH 7.4 and 150 mM KCl was used in both chambers. Proteoliposomes containing GIRK channels were fused to the bilayer membrane. 32 µM C8-$PIP_2$ and 2 mM $MgCl_2$ were added to the intracellular side of the chamber, and proteoliposomes containing Gβγ were fused to the membrane, activating GIRK channels. (**C**) GIRK1/4 hetero-tetramer currents recorded in the lipid bilayer. Membrane potential was held at 0 mV, and 10 mV voltage steps from -80 mV to 80 mV were applied. (**D**) GIRK4 homo-tetramer currents recorded in the lipid bilayer.

lipid molecules in the membrane (*Wang et al., 2016*). In subsequent graphs, Gβγ concentration is quantified as Ni-NTA-lipid mole fraction, but for accounting purposes the stoichiometry of sGβγ-$His_{10}$ to Ni-NTA-lipid is 1:3 (i.e. a single sGβγ-$His_{10}$ molecule binds to 3 Ni-NTA-lipid molecules), meaning the actual sGβγ-$His_{10}$ density on the membrane is one third the density of Ni-NTA-lipid (*Wang et al., 2016*). In order to compare currents from different membranes that generally contain different numbers of GIRK channels, at the end of each experiment proteolipsomes containing lipid-anchored Gβγ were fused to the membrane to maximally activate the GIRK channels in the membrane (*Figure 3B*). Current activated at a specific Gβγ concentration (determined by the density of Ni-NTA-lipids) is referred to as normalized current.

*Figure 4A* shows normalized GIRK4 current as a function of Gβγ concentration at 0 mM, 8 mM, and 32 mM $Na^+$ (*Figure 4A*). GIRK4 current increases as a sigmoid-shaped function, and $Na^+$ concentration has a prominent effect on Gβγ activation. Specifically, $Na^+$ increases GIRK4 current at all Gβγ concentrations, but notably, the increase is relatively largest at low Gβγ concentrations where, for example, at 0.001 Ni-NTA mole fraction 32 mM $Na^+$ increases normalized current almost 20-fold, from 0.018 to 0.34. These data suggest that GIRK4 is similar to the neuronal GIRK channel, GIRK2, in its response to Gβγ and $Na^+$ (*Wang et al., 2016*). We therefore applied the same equilibrium gating model used to analyze GIRK2 (*Wang et al., 2016*). The model has 25 states of ligand occupancy,

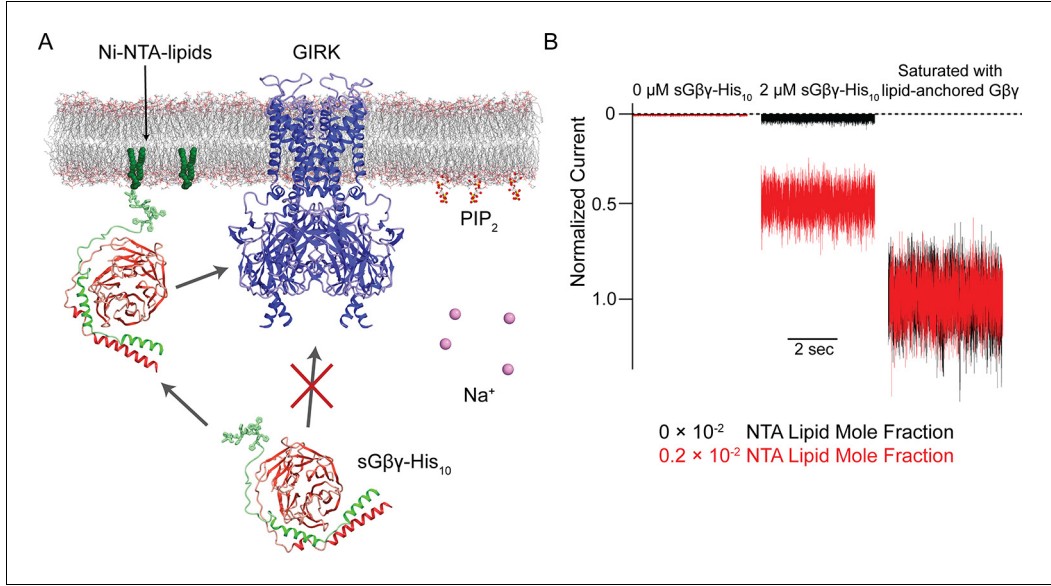

**Figure 3.** Schematic of the $Na^+$ and $G\beta\gamma$ titration using Ni-NTA-lipids. (**A**) GIRK channels were incorporated into the membrane containing a known concentration of Ni-NTA-lipids. 32 μM C8-PIP$_2$ and 2 μM sG$\beta\gamma$-His$_{10}$ were added to the intracellular side of the membrane. Free sG$\beta\gamma$-His$_{10}$ does not activate GIRK channels at the concentration applied, while Ni-NTA-lipids-bound sG$\beta\gamma$-His$_{10}$ mimics lipid-anchored G$\beta\gamma$ and activates GIRK channels. Known concentrations of $Na^+$ were subsequently added to study the effect of $Na^+$ concentration on GIRK channel activity in the presence of known concentrations of G$\beta\gamma$ in the membrane. (**B**) Left and center traces show normalized GIRK4 currents before and after application of 2 μM sG$\beta\gamma$-His$_{10}$ in the presence of 0 (black) or 0.002 (red) mole fraction of Ni-NTA-lipids in the membrane. At the end of each experiment, currents were fully activated by lipid-anchored G$\beta\gamma$ (right signals).

corresponding to 0 to 4 of each ligand, G$\beta\gamma$ and $Na^+$, as illustrated (*Figure 4D*). Parameters in the model include an equilibrium dissociation constant $K_{db}$ and cooperativity factor $b$ for G$\beta\gamma$ binding, an equilibrium dissociation constant $K_{dn}$ for $Na^+$ binding (the cooperativity factor for $Na^+$ binding is 1), a factor $\eta$ for the effect that G$\beta\gamma$ and $Na^+$ have on each other's affinity, and a term $\theta$ relating conductivity to ligand occupancy (*Table 1*). The model adequately represents the data with values for the parameters given (*Table 1*). The errors on values for equilibrium dissociation constants and cooperativity factors are larger than those determined for GIRK2 (*Wang et al., 2016*) because the data set on GIRK4 is smaller. However, the overall conclusion is that GIRK4 is very similar to GIRK2. Through model analysis the data support three conclusions: that 4 G$\beta\gamma$ molecules are required to open the channel (the model yields higher residuals with less than 4), that G$\beta\gamma$ binds cooperatively to GIRK4, and that $Na^+$ exerts its major effect by increasing the G$\beta\gamma$ affinity for the channel.

*Figure 4B* shows corresponding data for the GIRK1/4 channel. At all $Na^+$ concentrations – even in the absence of $Na^+$ – the response of the GIRK1/4 channel to G$\beta\gamma$ is similar to the GIRK4 channel at higher $Na^+$ concentrations. Thus, the GIRK1/4 hetero-tetramer channel, compared to the GIRK4 homo-tetramer channel, behaves to a first approximation as if it remains permanently stuck in a $Na^+$-activated state. That this influence of the GIRK1 subunit is related to its $Na^+$ binding site is supported by the mutation N217D, which converts the GIRK1 $Na^+$ binding locus to be more like that of GIRK4 by restoring its $Na^+$ sensitivity to the hetero-tetramer (*Figure 4C*) (*Ho and Murrell-Lagnado, 1999*). To test the idea that Asn217 in GIRK1 mimics a $Na^+$-bound Asp we fit the GIRK1/4 data to the same model used for the GIRK4 channel, but imposed the condition that two of the four sites are "permanently occupied" by $Na^+$, with the underlying idea that the two permanently occupied sites represent the GIRK1 subunits. This condition means GIRK1/4 is described by 15 states of ligand occupancy corresponding to 0 to 4 G$\beta\gamma$ and 0 to 2 $Na^+$. The model encodes this by collapsing the 0, 1 and 2 $Na^+$-occupied states of the GIRK4 model into a single state with affinity of G$\beta\gamma$ equal to $K_{db} \eta^2$ (*Table 1*). This model describes the data for the GIRK1/4 channel accurately with numerical values for $K_{db}$, $K_{dn}$, $b$ and $\eta$ that are indistinguishable from those for the GIRK4 model (*Table 1*).

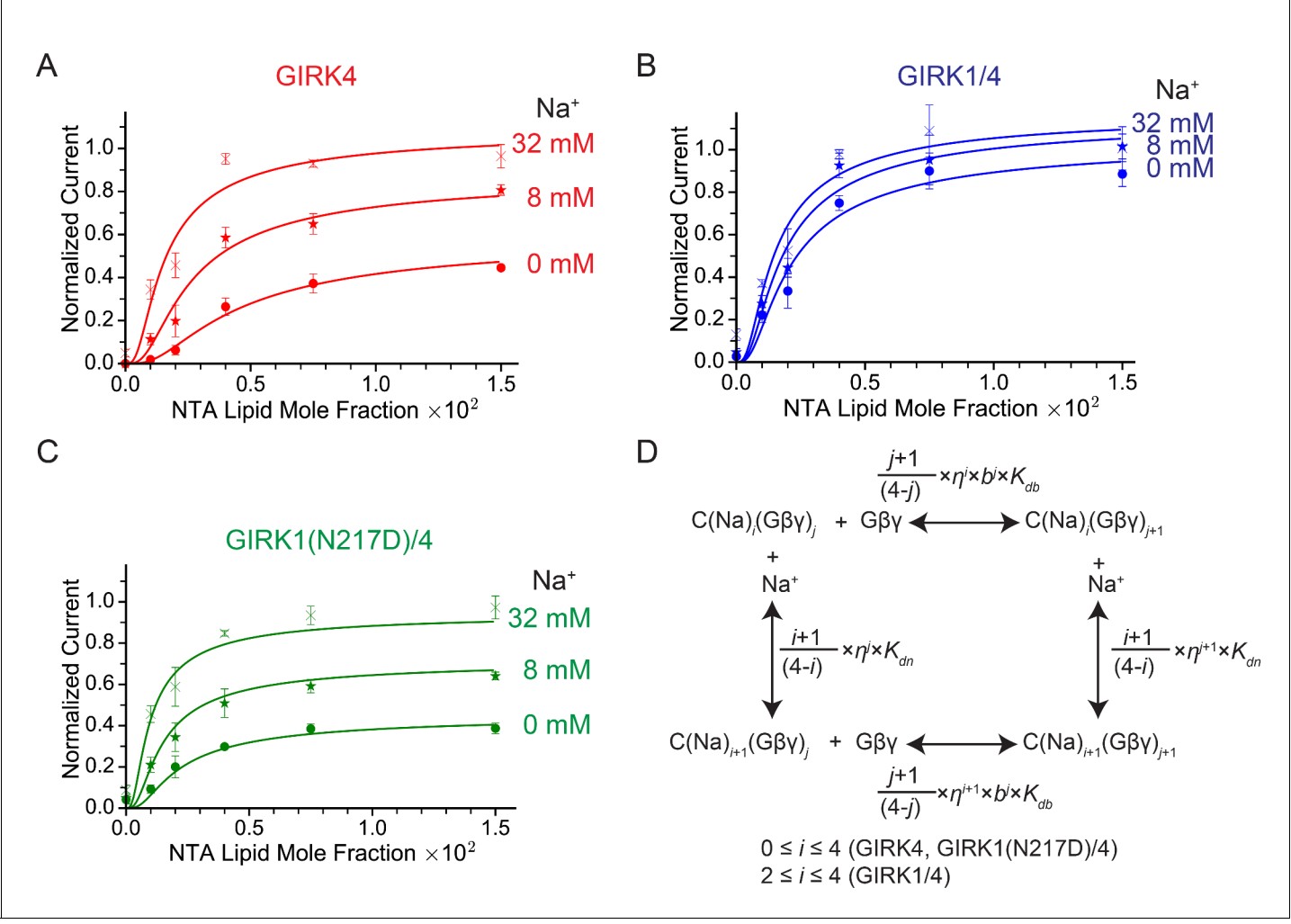

**Figure 4.** GIRK channel activity as a function of $Na^+$ and $G\beta\gamma$ concentrations. (**A**), (**B**), and (**C**) Plots of activity of GIRK4 homo-tetramers (**A**), GIRK1/4 hetero-tetramers (**B**), and GIRK1(N217D)/4 hetero-tetramers (**C**) versus Ni-NTA-lipid mole fraction in the membrane at different $Na^+$ concentrations. The same buffer (10 mM potassium phosphate pH 8.2, 150 mM KCl) was used in both chambers, and voltage across the lipid bilayer was held at -50 mV. GIRK proteoliposomes were fused to the bilayer membrane containing a known concentration of Ni-NTA-lipids. 2 mM $MgCl_2$ and 32 µM C8-$PIP_2$ were added to one side of the bilayer chamber and then 2 µM $sG\beta\gamma$-$His_{10}$ was added to the same side of the chamber to activate GIRK channels. 8 mM and 32 mM $Na^+$ were added to further activate GIRK channels. At the end of each experiment, channels were fully activated by fusing proteoliposomes containing lipid-anchored $G\beta\gamma$ and currents were normalized to the fully activated current (mean ± SEM, n = 3 bilayers). The equilibrium model (**D**) was used to fit the data (solid curves). $K_{db}$: Equilibrium dissociation constant between $G\beta\gamma$ and ligand-free GIRK. $K_{dn}$: Equilibrium dissociation constant between $Na^+$ and ligand-free GIRK (mM). $b$: Cooperativity factor for $G\beta\gamma$ binding. $\eta$: Cross-cooperativity factor between $G\beta\gamma$ and $Na^+$ binding. $i$: The number of $Na^+$ ions bound to GIRK. For GIRK1/4 hetero-tetramers, the range of $i$ was restricted to the range 2 to 4. $j$: The number of $G\beta\gamma$ bound to GIRK.

Thus, the properties of the GIRK1/4 channel are consistent quantitatively with the GIRK1 subunits functioning as if they are GIRK4 subunits with $Na^+$ ions permanently bound to them.

## Intracellular $Na^+$ does not significantly activate cardiac GIRK channels

In *Figure 5* we ask how does intracellular $Na^+$ affect GPCR-stimulated GIRK currents in mouse embryonic stem cell (mESC)-derived cardiac pacemaker cells. Whole-cell voltage clamp recordings show acetylcholine-activated $K^+$ currents that are inhibited by tertiapin-Q (TPNQ), a specific GIRK channel blocker (*Figure 5A*). Such recordings were performed with 22 different cells with intracellular solutions containing either 0 mM or 30 mM $Na^+$. Pacemaker cells showed an average of 32 ± 4 pA of acetylcholine-activated $K^+$ current in 0 mM $Na^+$ and 47 ± 6 pA in 30 mM $Na^+$ (*Figure 5B*).

**Table 1.** The fitting parameters for the Na$^+$ and Gβγ titration. $K_{db}$: Equilibrium dissociation constant for Gβγ in equilibrium with ligand-free GIRK. $K_{dn}$: Equilibrium dissociation constant for Na$^+$ binding to ligand-free GIRK (mM). $b$: Cooperativity factor for Gβγ binding. $\eta$: Cross-cooperativity factor between Gβγ and Na$^+$ binding. $\theta_{i,j}$: Normalized activity of $i$-Na$^+$ and $j$-Gβγ-bound GIRK. $R^2$: Adjusted R-squared. For fitting to GIRK1(N217D)/4 hetero-tetramers, $b$ and $\theta_{4,4}$ were fixed to the same parameters as GIRK1/4 hetero-tetramers.

| | GIRK4 | GIRK1/4 | GIRK1(N217D)/4 |
|---|---|---|---|
| $K_{db}$ | 0.004 ± 0.005 | 0.004 ± 0.006 | 0.0024 ± 0.0004 |
| $kK_{dn}$(mM) | 50 ± 40 | 50 ± 300 | 50 ± 30 |
| $b$ | 0.6 ± 0.3 | 0.6 ± 0.3 | 0.6 |
| $\eta$ | 0.7 ± 0.1 | 0.8 ± 0.4 | 0.71 ± 0.08 |
| $\theta_{0,4}$ | 0.6 ± 0.1 | - | 0.45 ± 0.04 |
| $\theta_{1,4}$ | $\theta_{0,4} + (\theta_{4,4} - \theta_{0,4}) \times 1/4$ | - | $\theta_{0,4} + (\theta_{4,4} - \theta_{0,4}) \times 1/4$ |
| $\theta_{2,4}$ | $\theta_{0,4} + (\theta_{4,4} - \theta_{0,4}) \times 2/4$ | 1.1 ± 0.1 | $\theta_{0,4} + (\theta_{4,4} - \theta_{0,4}) \times 2/4$ |
| $\theta_{3,4}$ | $\theta_{0,4} + (\theta_{4,4} - \theta_{0,4}) \times 3/4$ | 1.2 ± 0.8 | $\theta_{0,4} + (\theta_{4,4} - \theta_{0,4}) \times 3/4$ |
| $\theta_{4,4}$ | 1.2 ± 0.1 | 1.1 ± 0.8 | 1.1 |
| $R^2$ | 0.96 | 0.93 | 0.97 |

We thus conclude that intracellular Na$^+$ has essentially no influence on GPCR-stimulated GIRK current in these mESC-derived cardiac cells. This observation is consistent with the data recorded in bilayers if the cardiac cells express predominantly GIRK1/4 hetero-tetramer channels, which are only weakly Na$^+$ sensitive, and not GIRK4 homo-tetramer channels, which are strongly Na$^+$ sensitive (*Figure 4*).

## Discussion

In cardiac cells two different subunits, GIRK1 and GIRK4, form G protein gated K$^+$ channels. Homo-tetramers of GIRK4 and hetero-tetramers of GIRK1 and GIRK4, GIRK1/4, form functional K$^+$ channels, while homo-tetramers of GIRK1 do not (*Krapavinsky et al., 1995*; *Hedin et al., 1996*). It is unclear to what extent GIRK4 homo-tetramers versus GIRK1/4 hetero-tetramers dominate in cardiac cells. It is also unclear to what extent the functional properties of these channels differ because it has not been possible to study GIRK1/4 channels in isolation, the reason being heterologous expression of both subunits naturally gives rise to a mixed population of homo- and hetero-tetramers. To overcome this problem we overexpressed and purified GIRK1/4 hetero-tetramers using sequential affinity columns and also expressed and purified GIRK4 homo-tetramers for comparative analysis. The composition of GIRK1/4 hetero-tetramers is reported to consist mainly of two GIRK1 and two GIRK4 subunits (*Silverman et al., 1996*; *Corey et al., 1998*). In this study purified GIRK1/4 hetero-tetramers are also most likely composed of two GIRK1 subunits and two GIRK4 subunits, as estimated from SDS-PAGE of fractions from a gel filtration column (*Figure 2B*). However, we have no information on the arrangement of subunits within the tetramer either in cells or in our reconstitution experiments.

We observe that GIRK4 homo-tetramers and GIRK1/4 hetero-tetramers exhibit distinctly different properties with respect to their activation by Gβγ and Na$^+$. It had been shown that the GIRK1 subunit has a defective Na$^+$ site (*Ho and Murrell-Lagnado, 1999*), but the present study establishes the following new conclusions. First, that Na$^+$ binding to the GIRK4 subunit increases affinity for Gβγ This effect is encoded in the model by the Gβγ-Na$^+$ cross interaction term $\eta$. Second, the GIRK1 subunit behaves similarly to the GIRK4 subunit with Na$^+$ permanently bound. Thus, while the GIRK1 subunit is unable to bind Na$^+$, it causes the channel to have high affinity for Gβγ even in the absence of Na$^+$. This effect is encoded in the model by enforcing permanent Na$^+$ occupancy on the GIRK1 subunits. Taken together, these properties account for the functional differences we observe between GIRK4 and GIRK1/4 channels. GIRK4 channels are less sensitive to G protein stimulation at low Na$^+$ concentrations (Gβγ binds with lower affinity) and more sensitive at high Na$^+$ concentrations (Gβγ

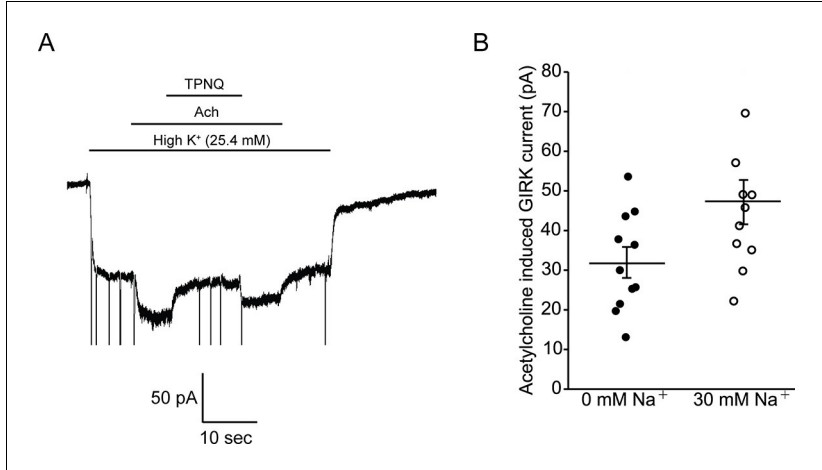

**Figure 5.** Intracellular $Na^+$ does not significantly activate cardiac GIRK channels. (**A**) Whole-cell voltage clamp recordings on mESC-derived pacemaker cells. Membrane potential was held at -80 mV and the extracellular solution was exchanged to high potassium buffer (25.4 mM KCl) as indicated above the signal. 10 µM acetylcholine (Ach) was then applied to activate GIRK channels and 100 nM tertiapin Q (TPNQ) was next applied to block cardiac GIRK currents. Acetylcholine-activated GIRK currents were measured by subtracting signals before and after acetylcholine application. (**B**) Acetylcholine induced GIRK currents were measured with the pipette solution containing 0 mM $Na^+$ or 30 mM $Na^+$. Eleven recordings were performed with each pipette and average value was calculated (mean ± SEM, n = 11 cells).

binds with higher affinity). GIRK1/4 channels on the other hand are very sensitive to Gβγ stimulation at both low and high $Na^+$ concentrations (Gβγ binds with high affinity independent of $Na^+$ concentration).

We also find that GPCR-stimulated GIRK currents in mESC-derived cardiac pacemaker cells are nearly independent of intracellular $Na^+$ concentration. Based on a comparison of these cellular data to the properties of isolated GIRK4 and GIRK1/4 channels in planar lipid bilayers, we conclude that GIRK channels in mESC-derived cardiac channels most likely are predominantly GIRK1/4 hetero-tetramers. In an accompanying paper we report that GIRK channels in mouse dopamine neurons are very sensitive to intracellular $Na^+$: in experiments analogous to those in *Figure 5B*, eight fold amplification of GPCR-stimulated GIRK currents was observed (*Wang et al., 2016*). This degree of $Na^+$ sensitivity is consistent with neurons expressing GIRK2 homo-tetramers. GIRK2, like GIRK4, encodes an intact $Na^+$ activation site.

Our findings lead us to conclude that the GIRK1 subunit in a GIRK1/4 hetero-tetramer renders the channel relatively insensitive to $Na^+$ but permanently in a state of high responsiveness to GPCR stimulation. We can only speculate as to why two kinds of GIRK channels exist, ones whose G protein sensitivity is regulated by intracellular $Na^+$ (i.e. homo-tetramer GIRK4 or GIRK2 channels) and ones whose G protein sensitivity is not much regulated by $Na^+$ but is always near maximum (i.e. hetero-tetramer GIRK1/4 channels). In neurons, intracellular $Na^+$ concentration increases during excitation because $Na^+$ enters the cell through $Na^+$ channels during action potentials and through glutamate receptor ion channels in response to excitatory neurotransmitters (*Lasser-Ross and Ross, 1992*). GIRK2 channels silence neurons in response to inhibitory neurotransmitters, which act through inhibitory GPCRs. The GIRK2 regulation by $Na^+$ provides a way to modulate the inhibitory response according to the activity level. Such modulation would seem beneficial to a neuron that exhibits a wide range of electrical activity from near silent to high frequency spiking. Cardiac cells on the other hand appear to exhibit less activity-dependent variation in levels of intracellular $Na^+$ (*Harrison et al., 1992*). Thus, it seems reasonable that GIRK1/4 channels do not exhibit high $Na^+$ sensitivity, but instead exhibit a permanent state of cholinergic responsiveness (*Ito et al., 1994*).

## Materials and methods

### Expression and purification

Human full-length GIRK1 and GIRK4 were cloned into pEG BacMam (*Goehring et al., 2014*). At the C-terminus of the GIRK1 construct, a PreScission protease cleavage site, an enhanced green fluorescent protein (eGFP), and a deca-histidine tag were attached for purification. A 1D4 peptide tag was used instead of a deca-histidine tag for the GIRK4 construct. These constructs were used for fluorescent size exclusion chromatography (FSEC), and overexpression and protein purification. For FSEC, HEK293T cells were transiently transfected with GIRK1-His$_{10}$-pEG BacMam or GIRK4-1D4-pEG BacMam, and incubated at 37°C for 48 hr. Cells were solubilized in 50 mM HEPES-KOH (pH 7.35), 150 mM KCl, 4% (w/v) n-decyl-η-D-maltopyranoside (DM), and a protease inhibitor cocktail (1 mM PMSF, 0.1 mg/mL trypsin inhibitor, 1 µg/mL pepstatin, 1 µg/mL leupeptin, and 1 mM benzamidine). Lysed cells were centrifuged and supernatant was run on FSEC (Superose 6 10/300 GL). For overexpression and protein purification, HEK293S GnTl⁻ cells were grown in suspension, transduced with P3 BacMam virus of the GIRK1-His and the GIRK4-1D4 in 1:1 ratio, and incubated at 37°C 8-12 hr post-transduction, 10 mM sodium butyrate was added to the culture and cells were harvested 60 hr post-transduction. Cells were harvested by centrifugation, frozen in liquid N$_2$, and stored at -80°C until needed. Frozen cells were solubilized in 50 mM HEPES-KOH (pH 7.35), 150 mM KCl, 4% (w/v) DM, and protease inhibitor cocktail. 2 hr after solubilization, lysed cells were centrifuged and supernatant was incubated with the Talon metal affinity resin (Clontech Laboratories, Inc. Mountain View, CA) for 1 hr at 4°C with gentle mixing. The resin was washed in batch with 5 column volume (cv) of buffer A (50 mM HEPES-KOH [pH 7.0], 150 mM KCl, 0.4% [w/v] DM), then loaded onto a column and further washed with 5 cv buffer A + 10 mM imidazole. The column was then eluted with buffer A + 200 mM imidazole. The peak fraction was collected and incubated with the 1D4 affinity resin for 1 hr at 4°C with gentle mixing. The resin was loaded onto a column and washed with buffer A. 5 mM DTT and 1 mM EDTA were added and eGFP and affinity tags were cut with PreScission protease overnight at 4°C. The cleaved protein was then concentrated to run on a Superose 6 10/300 GL gel filtration column in 20 mM Tris-HCl (pH 7.5), 150 mM KCl, 0.2% (w/v) DM, 20 mM DTT, and 1 mM EDTA. GIRK1(N217D)/GIRK4 hetero-tetramers were purified using the same procedure. GIRK4 homo-tetramers were purified with a similar procedure. In brief, GIRK4 homo-tetramers were expressed in HEK293S GnTl⁻ cells and purified using the 1D4 affinity chromatography and size-exclusion chromatography. Human lipid-anchored Gβγ and soluble deca-histidine tagged Gβγ were purified as described (*Wang et al., 2016*).

### Proteoliposome reconstitution

All lipids were purchased from Avanti Polar Lipids (Alabaster, AL). Proteoliposomes were reconstituted as described (*Wang et al., 2016*). In brief, 20 mg/mL of the lipid mixture (3:1 [wt:wt] = 1-palmitoyl-2-oleyl-sn-glycero-3-phosphoethanolamine [POPE]: 1-palmitoyl-2-oleyl-sn-glycero-3-phospho-[1'-rac-glycerol] [POPG]) was dispersed by sonication and solubilized with 20 mM DM. In the Na⁺ and Gβγ titration experiment, 0–0.015 (mole fraction) of 1,2-dioleoyl-sn-glycero-3-[(N-(5-amino-1-carboxypentyl)iminodiacetic acid)succinyl] (nickel salt) (DOGS-NTA) were further added to the lipid mixture.

Purified GIRK channels were combined with the lipid mixture in a GIRK: lipid (wt:wt) ratio of 1:10. Protein-lipid mixtures were then diluted into the reconstitution buffer (10 mM potassium phosphate [pH 7.4], 150 mM KCl, 1 mM EDTA, and 3 mM DTT) to 1 mg/mL (GIRK) and 10 mg/mL (lipid mixture). Detergent was removed by dialysis against the reconstitution buffer at 4°C for 4 days. Gβγ proteoliposomes were prepared as described (*Wang et al., 2014*).

### Planar lipid bilayer recordings

Bilayer experiments were performed as described (*Wang et al., 2016*). In brief, 20 mg/mL of a lipid solution in decane composed of 2:1:1 (wt:wt:wt) of 1,2-dioleoyl-sn-glycero-3-phosphoetanolamine (DOPE), 1-palmitoyl-2-oleyl-sn-glycero-3-phosphocholine (POPC), and 1-palmitoyl-2-oleoyl-sn-glycero-3-phospho-L-serine (POPS) was painted over a ~120 µm hole on a piece of transparency film. For the Na⁺ and Gβγ titration experiments, 0–0.015 (mole fraction) of DGS-NTA was added to a lipid solution in decane composed of 1:1 (wt:wt) of DOPE and POPC, and the lipid mixture was painted

over a transparency film. The same buffer (10 mM potassium phosphate pH 7.4 or pH 8.2 for the $Na^+$ and $G\beta\gamma$ titration experiment, 150 mM KCl) was used in both chambers. Voltage across the lipid bilayer was clamped with an Axopatch 200B amplifier (Molecular Devices, Sunnyvale, CA) in whole-cell mode. The analog current signal was low-pass filtered at 1 kHz (Bessel) and digitized at 20 kHz with Digidata 1322A or Digidata 1440A digitizer (Molecular Devices). Digitized data were recorded on a computer using the software pClamp (Molecular Devices). Measurements were replicated on three membranes, and average and SEM values were calculated for each data point.

## Whole-cell voltage clamp recordings on HEK cells

Human M2R was cloned into the pIRES-mCherry vector for mammalian cell expression. HEK293T cells were transiently transfected with GIRK1-His$_{10}$-pEG BacMam or GIRK4-1D4-pEG BacMam, and M2R-pIRES-mCherry were incubated at 37°C for 24–36 hr. Cells were dissociated and plated on PDL-coated glass coverslips for electrophysiological recordings. Whole-cell voltage clamp recordings were performed with an Axopatch 200 B amplifier in whole-cell mode. The analog current signal was low-pass filtered at 1 kHz (Bessel) and digitized at 20 kHz with a Digidata 1440 A digitizer. Digitized data were recorded on a computer using the software pClamp. Patch electrodes (resistance 2.0–4.0 MΩ) were pulled on Sutter P-97 puller (Sutter Instrument Company, Novato, CA) from 1.5 mm outer diameter filamented borosilicate glass. Membrane potential was held at -80 mV throughout the experiments, and the extracellular solution was exchanged with local perfusion with a 100 μM diameter perfusion pencil positioned beside the cell. The low potassium extracellular solution contained (in mM): 150 NaCl, 3 KCl, 2.5 CaCl$_2$, 1 MgCl$_2$, 10 D-glucose, 10 HEPES-NaOH (pH 7.4) (~320 mOsm). The high potassium extracellular solution contained (in mM): 53 NaCl, 100 KCl, 2.5 CaCl$_2$, 1 MgCl$_2$, 10 D-glucose, 10 HEPES-NaOH (pH7.4) (~311 mOsm) and 10 μM acetylcholine was added. The pipette solution contained (in mM): 150 KCl, 2 MgCl$_2$, 5 EGTA-K, 10 HEPES-KOH (pH7.4) (~310 mOsm).

## Differentiation of mouse embryonic stem cells

W4 (129sv) ES cell line was cultured in 2i/LIF medium (*Auerbach et al., 2000*; *Ying et al., 2008*). All ES culture reagents were purchased from Thermo Fisher Scientific (Waltham, MA) except for 2i and LIF (EMD Millipore, Billerica, MA). ESCs were differentiated into spontaneously beating cardiomyocytes with the hanging drop method (*Maltsev et al., 1993*). Embryoid bodies (EBs) were formed in hanging drops of ~20 μL from ~1000 cells in differentiation medium (GMEM, 10% ES-FBS, 2 mM L-glutamine, 1 mM sodium pyruvate, 1x non-essential amino acids, 0.1 mM 2-mercaptoethanol) and were cultivated in hanging drops for 5 days. Single EBs were transferred into gelatin-coated 48-well plates, and observed daily. Spontaneously contracting EBs were observed around day 8.

## Preparation of single pacemaker cells

Contracting regions of day 16–18 EBs were dissected with micro knives, and collected into the solution containing (in mM): 120 NaCl, 5.4 KCl, 5 MgSO$_4$, 20 Glucose, 10 HEPES-NaOH (pH 6.9), 20 Taurine. Collected cells were digested with 50 μM CaCl$_2$ + 1 mg/mL type-II collagenase (Sigma-Aldrich, St. Louis, MO) for 30 min, and plated on 12 mm PDL-coated glass coverslips. Electrophysiological recordings were performed 24–48 hr after the dissociation. On average approximately three beating cells were identified per coverslip.

## Whole-cell voltage clamp recordings on pacemaker cells

Whole-cell voltage clamp recordings were performed with the same setup, pipettes, and perfusion system as described above. After the whole-cell configuration was formed, membrane potential was held at -80 mV in low potassium extracellular solution for about 3 min to equilibrate the intracellular solution with the pipette solution. The low potassium extracellular solution contained (in mM): 140 NaCl, 5.4 KCl, 2 CaCl$_2$, 1 MgCl$_2$, 10 D-glucose, 10 HEPES-NaOH (pH 7.4) (~300 mOsm). The high potassium extracellular solution contained (in mM): 120 NaCl, 25.4 KCl, 2 CaCl$_2$, 1 MgCl$_2$, 10 D-glucose, 10 HEPES-NaOH (pH7.4) and 10 μM acetylcholine and 100 nM TPNQ were added (~300 mOsm). 0 mM $Na^+$ pipette solution contained (in mM): 100 K-PO$_4$, 30 NMDG-Cl, 10 EGTA-K, 2 MgCl$_2$, 10 HEPES-KOH (pH7.0) (~315 mOsm). 30 mM $Na^+$ pipette solution contained (in mM):

100 K-PO$_4$, 30 NaCl, 10 EGTA-K, 2 MgCl$_2$, 10 HEPES-KOH (pH7.0) ($\sim$315 mOsm). 0.25 mM Na-GTP and 3 mM Mg-ATP were supplemented to pipette solutions just before the experiments.

## Acknowledgements

We thank Yi Chun Hsiung for assistance with mammalian and insect cell culture, members of the MacKinnon laboratory for helpful discussions and Eunyong Park for advice on the manuscript. This work was supported in part by NIHGM43949. RM is an investigator in the Howard Hughes Medical Institute.

## Additional information

### Funding

| Funder | Grant reference number | Author |
| --- | --- | --- |
| National Institutes of Health | NIHGM43949 | Roderick MacKinnon |
| Howard Hughes Medical Institute | | Roderick MacKinnon |

The funders had no role in study design, data collection and interpretation, or the decision to submit the work for publication.

### Author contributions

KKT, Designed the study, Collected electrophysiology data in planar lipid membranes, Performed the mESC differentiation and carried out electrophysiology recordings on pacemaker cells, Performed the equilibrium modeling, Analyzed data, Wrote the paper; WW, Taught and helped to collect electrophysiology data in planar lipid membranes, Performed the equilibrium modeling, Analyzed data, Input to write the paper; RM, Designed the study, Helped to perform the equilibrium modeling, Analyzed data, Wrote the paper

### Author ORCIDs

Roderick MacKinnon, http://orcid.org/0000-0001-7605-4679

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
