## [Decision Letter]

Thank you for submitting your article "The GIRK1 subunit potentiates G protein activation of cardiac GIRK1/4 hetero-tetramers" for consideration by *eLife*. Your article has been reviewed by three peer reviewers, and the evaluation has been overseen by a Reviewing Editor and Richard Aldrich as the Senior Editor.

The reviewers have discussed the reviews with one another and the Reviewing Editor has drafted this decision to help you prepare a revised submission.

The following individuals involved in the review process of your submission have agreed to reveal their identity: Kenton Swartz (Reviewing editor); Chris Miller, David Clapham and Don Hilgemann (peer reviewers).

Summary:

There are 4 inwardly rectifying K channel subunits that form tetrameric channels that are activated by G protein β/γ (Gβγ) subunits, GIRK1-4. Their combination into tetramers is not random – a long term question in the field is the different regulatory or functional consequences of heteromerization. Touhara et al. show that intracellular Na^+^ regulates homotetrameric GIRK4 channels sensitivity to Gβγ. In contrast, GIRK1 lacks an optimal Na binding site and GIRK1/4 channel Gβγ sensitivity is near its maximum. In other words, GIRK1 subunits may be evolved to reduce Na sensitivity in some tissues, such as heart (GIRK1 + GIRK4), while Na sensitive subunits (GIRK4 homomers) are adapted to regulate neuronal firing as sodium accumulates during bursts of action potentials. The authors take an innovative and solid approach to answering this question and do so beautifully in a reconstituted system where stoichiometry and composition of subunits can be controlled. They do this through use of distinct tags for GIRK1 and GIRK4 subunits that enable their purification, and then controlling the Gβγ incorporation into the membrane by tagging Gβγ with His_10_ and Ni-NTA tagged lipids. This enables them to quantify the relation between NTA mole fraction of membrane bound Gβγ and GIRK4 vs GIRK1/4 current. They then summarize this in a multi- state model that incorporates Gβγ and Na. The main finding is that Na binding to the GIRK4 subunit increases the affinity for Gβγ, while Gβγ does not require Na binding for its high affinity for Gβγ. The data are outstanding and the interpretation convincing. This manuscript harvests the gold mined in its methodological partner-paper that sets the stage for a quantitative understanding of Gβγ-activated, Na^+^-modulated GRIK channel behavior, by systematically varying Gβγ 'concentration' (i.e., membrane surface density). The manuscript is thorough, in not only purifying and quantifying reconstituted GIRK heterotets, or in comparing Gβγ activation of GRK4 homotetramers with GRK1/GRIK4 channels and the striking difference of Na^+^ modulation accomplishments that in themselves would be an exciting breakthrough, but in also linking this new information to GIRK behavior in cardiac and neuronal contexts.

Requested revisions:

1) The parameters *K_db_*and *K_dn_* are described as "equilibrium constants". After several readings, I managed to conclude (a bit shakily) that these refer to dissociation constants rather than binding constants. Why don't you say this up front?

2) The parameters of Table 1 have no units associated with them.

3) In the second paragraph of the subsection “The Na^+^-insensitive GIRK1 subunit potentiates Gβγ activation of GIRK1/4 hetero tetramers”: Instead of just tersely telling us that the model represents the data well, couldn't you offer the reader a bit more narration about the values of the dissociation constants and the coop factors (beyond telling us to go look at the table), either here or in the second paragraph of the Discussion? For instance, how much extra free energy of binding do the *η* and *b* factors represent?

4) The authors consider that the Na-dependent activation of GIRKs will modulate firing rate in dependent on Na accumulation that occurs as frequency of firing increases. They speculate that cell expression of different GIRK isoforms will modulate firing in different ways. While this seems obvious, it is really only the case in a simple way within cell types with very similar electrophysiological properties. It may not be the case when comparing across cell types with different expression of ion channels and transporters. Consider the cardiac SA node: It is not at all obvious that cytoplasmic Na will accumulate with increased frequency in S/A node cells. The upstroke is made by Ca influx, and Na influx is occurring primarily during the diastolic period through CNG channels (probably). Thus, it is more likely that cytoplasmic Na will decrease with increase of frequency, given this situation, thereby creating a very different regulatory situation than in neurons using Nav channels to make APs. Admittedly, this conclusion may be altered if Na/Ca exchangers generate the majority of Na influx in SA node cells, but most recent work does not support this possibility. We believe that these considerations can be used by the authors to improve the clarity and future relevance of their discussion.

---

## [Author Response]

*Requested revisions: 1) The parameters K_db_ and K_dn_ are described as "equilibrium constants". After several readings, I managed to conclude (a bit shakily) that these refer to dissociation constants rather than binding constants. Why don't you say this up front?* We have changed them to “equilibrium dissociation constant”.

2) The parameters of Table 1 have no units associated with them.

We have put the unit for *K_dn_* in Table 1.

3) In the second paragraph of the subsection “The Na^+^-insensitive GIRK1 subunit potentiates Gβγ activation of GIRK1/4 hetero tetramers”: Instead of just tersely telling us that the model represents the data well, couldn't you offer the reader a bit more narration about the values of the dissociation constants and the coop factors (beyond telling us to go look at the table), either here or in the second paragraph of the Discussion? For instance, how much extra free energy of binding do the *η* and *b* factors represent?

We have expanded the discussion on dissociation constants and cooperativity factors in the second paragraph of the subsection “The Na^+^-insensitive GIRK1 subunit potentiates Gβγ activation of GIRK1/4 hetero94 tetramers”.

4) The authors consider that the Na-dependent activation of GIRKs will modulate firing rate in dependent on Na accumulation that occurs as frequency of firing increases. They speculate that cell expression of different GIRK isoforms will modulate firing in different ways. While this seems obvious, it is really only the case in a simple way within cell types with very similar electrophysiological properties. It may not be the case when comparing across cell types with different expression of ion channels and transporters. Consider the cardiac S/A node: It is not at all obvious that cytoplasmic Na will accumulate with increased frequency in S/A node cells. The upstroke is made by Ca influx, and Na influx is occurring primarily during the diastolic period through CNG channels (probably). Thus, it is more likely that cytoplasmic Na will decrease with increase of frequency, given this situation, thereby creating a very different regulatory situation than in neurons using Nav channels to make APs. Admittedly, this conclusion may be altered if Na/Ca exchangers generate the majority of Na influx in SA node cells, but most recent work does not support this possibility. We believe that these considerations can be used by the authors to improve the clarity and future relevance of their discussion.

As you point out, cardiac cells appear to keep the intracellular Na^+^ concentration relatively stable. We have added a reference to cite this work, and modified the discussion of this point.